# Cellular Response of Immune Cells in the Upper Respiratory Tract After Treatment with Cold Atmospheric Plasma In Vitro

**DOI:** 10.3390/ijms26010255

**Published:** 2024-12-30

**Authors:** Leonardo Zamorano Reichold, Michael Gruber, Petra Unger, Tim Maisch, Regina Lindner, Lisa Gebhardt, Robert Schober, Sigrid Karrer, Stephanie Arndt

**Affiliations:** 1Department of Dermatology, University Medical Center Regensburg, 93053 Regensburg, Germany; leonardo.zamorano-reichold@stud.uni-regensburg.de (L.Z.R.); petra.unger@ukr.de (P.U.); tim.maisch@ukr.de (T.M.); sigrid.karrer@ukr.de (S.K.); 2Department of Anesthesiology, University Medical Center Regensburg, 93053 Regensburg, Germany; michael.gruber@ukr.de (M.G.); regina.lindner@ukr.de (R.L.); 3Terraplasma Medical GmbH, 85748 Garching, Germany; lisa.gebhardt@terraplasma-medical.com (L.G.); schober@terraplasma.com (R.S.)

**Keywords:** cold atmospheric plasma (CAP), upper respiratory tract (URT), polymorphonuclear neutrophils (PMNs), cell migration, reactive oxygen/nitrogen species (RONS), NETosis, immunomodulation, ventilator-associated pneumonia (VAP)

## Abstract

Cold atmospheric plasma (CAP) has antimicrobial properties and is also known to stimulate the immune system. These properties could be useful for the development of a novel therapeutic or preventive strategy against respiratory infections in the upper respiratory tract (URT) such as ventilator-associated pneumonia (VAP) without inducing an immune overreaction. This study investigated the cellular responses of polymorphonuclear neutrophils (PMNs) after exposure to CAP in a three-dimensional (3D) model of the URT. In vitro experiments were conducted using PMNs isolated from human blood to assess cell migration, intracellular production of reactive oxygen species (ROS), NETosis, surface marker expression (CD11b, CD62L, and CD66b), and cell death with live cell imaging and flow cytometry. CAP was applied for 5 min using two distinct modalities: pressurized air plasma with a plasma intensive care (PIC) device and nebulized air plasma (NP) with a new humidity resistent surface microdischarge (SMD) plasma source, both developed by Terraplasma Medical GmbH. There were no significant signs of cell damage or overstimulation with either device under the conditions tested. However, the NP device caused milder effects on PMN functionality compared to the PIC device, but also demonstrated reduced antibacterial efficacy and reactive oxygen/nitrogen species (RONS) production, as analyzed with colorimetric/fluorimetric assay kits. These findings highlight a trade-off between the two CAP modalities, each with distinct advantages and limitations. Further studies are necessary to investigate these effects in the clinical setting and evaluate the long-term safety and efficacy of CAP treatment in the URT.

## 1. Introduction

Cold atmospheric plasma (CAP) has gained considerable attention in recent years because of its broad spectrum of antimicrobial properties and its potential to modulate the immune response [1]. CAP is a partially ionized gas that contains a variety of components including electrons, ions, neutral atoms and molecules, ultraviolet light (UV), and various reactive oxygen and nitrogen species (RONS). Reactive oxygen species (ROS), e.g., singlet oxygen (^1^O_2_), superoxide anion (O_2−_), hydrogen peroxide (H_2_O_2_), ozone (O_3_), and hydroxyl radicals (•OH), are known for their ability to induce oxidative stress in cells [2,3,4]. These properties can cause apoptosis or necrosis in cancer cells and damage microbial cell membranes, proteins, and DNA, ultimately eradicating pathogens with no known development of resistance [5,6,7]. Reactive nitrogen species (RNS), which include nitric oxide (NO), nitrite (NO_2−_), and nitrate (NO_3−_), may also contribute to cellular signaling and have antimicrobial properties [8,9,10].

Initially developed for use in wound healing, CAP has demonstrated efficacy in reducing microbial load while promoting tissue regeneration [10,11,12,13]. However, the versatility of CAP lies in its ability to be applied in a variety of forms, including direct plasma jet, remote plasma application, and plasma-treated solutions, facilitating a wide range of medical applications beyond dermatology [3,14,15,16]. A key advancement in the medical application of CAP has been made by Terraplasma Medical GmbH (Garching, Germany)**,** which developed a method to apply CAP homogeneously through a tube system. Pressurized air plasma is generated by the designated plasma intensive care (PIC) device using the surface microdischarge (SMD)-based plasma care^®^ device (Terraplasma Medical GmbH, Garching, Germany) as published recently [17]. The PIC device allows CAP to reach areas directly within the respiratory tract that are typically inaccessible with conventional therapies.

A promising area of research lies in the potential of CAP as a novel therapeutic or preventive strategy for hospital-acquired infections, in particular ventilator-associated pneumonia (VAP), a serious and often life-threatening condition prevalent in intensive care units. VAP is defined as pneumonia that develops in patients who have been mechanically ventilated for at least 48 h [18,19,20,21]. It is associated with high mortality rates and an increased reliance on antibiotics, exacerbating the growing problem of antibiotic resistance [19,20,21]. In addition, the financial burden of VAP is significant, resulting in increased hospital costs due to longer hospitalization, additional treatments, and the intensive care required to manage such infections [19,20,22,23]. Previous studies have demonstrated the safe use of CAP in various eukaryotic cells both in vitro and in vivo experiments, as well as its strong antibacterial efficacy [10,17,24,25,26,27]. However, its application in sensitive areas, such as the respiratory tract, still requires thorough investigation to avoid potential side effects, particularly excessive immune responses or cytotoxicity caused by RONS production [28].

Using the PIC device and a three-dimensional (3D) model of the upper respiratory tract (URT), Karrer et al. (2024) conducted an in vitro safety study on the treatment of oral keratinocytes, human bronchial/tracheal epithelial cells, and human lung fibroblasts that showed that CAP does not significantly affect cell viability, morphology, apoptosis, DNA damage, or migration. However, immunomodulatory effects, such as the increased release of pro-inflammatory cytokines (MCP-1, IL-6, IL-8, TNF-α), were observed [17]. A study on polymorphonuclear neutrophils (PMNs) by Kupke et al. (2021) showed that CAP clearly affects PMNs, although without evidence of inducing an immune overreaction. However, their study was conducted with a different plasma source (plasma care^®^ device from Terraplasma Medical GmbH, Garching, Germany) developed for wound healing purposes [15]. Thus, despite promising evidence on the safety of CAP treatment, concerns remain about potential immune overreaction in the respiratory tract.

This study aims to bridge the gap between these two areas of research by treating PMNs with CAP in a 3D model of the URT. By combining the approaches of Karrer et al. (2024) and Kupke et al. (2021), the present study integrates both the cellular environment of the respiratory tract and the immune-modulating effects of CAP on PMNs as key immune cells. This study aims to provide a more comprehensive understanding of how CAP influences immune cells in the context of respiratory infections.

A key aspect of this research is the use and comparison of two distinct modalities of CAP generations to find out whether the effects on PMNs in the URT can be modulated on a per device basis. This was achieved by analyzing various device-specific cellular and molecular biological parameters (e.g., RONS generation, antibacterial efficacy, migration, NETosis, intracellular ROS production, surface antigen expression, and cell death). For this approach, the PIC device already employed in the study by Karrer et al. 2024 was used [17]. The PIC device and the setup for PMN treatments are described in the Materials and Methods section of the manuscript (Section 4.2) and depicted in Figure 1a. The PIC device was compared to a prototype of a newly developed humidity resistent SMD plasma source in combination with a nebulizer construction named nebulized air plasma (NP) device. A detailed description of the NP device and the complete setup is described in Section 4.2 and shown in Figure 1b. Both devices were converted for a comparable application in the 3D URT model (Figure 1c,d).

The addition of aerosols to CAP can significantly alter its composition and properties. Nebulized air contains water droplets and various gases, in particular oxygen and nitrogen. These additional molecules can interact with the reactive species in CAP, changing the formation of RONS. The addition of mist increases the moisture content of the plasma, which may influence chemical reactions, as water acts as a solvent for many biochemical processes and can also contribute to the formation of hydroxyl radicals (•OH) [29]. Changes in the chemical composition may also affect the biological effects of the plasma. For example, a higher concentration of RONS may increase antimicrobial activity or influence cellular behavior in therapeutic applications. Overall, the addition of aerosols creates a more complex chemical environment in CAP that can have both positive and negative effects on its medical applications.

The present study compares the two CAP treatment modalities, NP and PIC, on PMNs in the URT model to evaluate their efficacy and safety for future use in the treatment or prevention of respiratory infections such as VAP.

## 2. Results

To evaluate the efficacy and safety of CAP treatment in the URT, PMNs were isolated from whole blood and exposed to CAP using a 3D URT model that included two anatomical treatment sites: the “throat” and the “subglottis”. The study used two different CAP devices (NP and PIC), each with unique plasma modalities as described in detail in Section 4.2, and various control groups, including untreated controls (ctrl.), pressurized air controls (PA), and nebulized air controls (N).

Initially, the research focused on device-specific analyses to measure levels of reactive oxygen species (ROS), specifically hydrogen peroxide (H_2_O_2_), and reactive nitrogen species (RNS), such as nitrite (NO_2−_) and nitrate (NO_3−_). These measurements were made using colorimetric and fluorimetric assay kits on blood serum and Dulbecco’s Phosphate-Buffered Saline (DPBS) containing isolated PMNs, providing insight into the distribution of reactive oxygen and nitrogen species (RONS) in the fluids used in the URT model. Additionally, the antibacterial efficacy of the plasma sources was assessed at both the “throat” and “subglottis” positions in the URT model, using *E. coli* as a representative bacterial organism.

The second part of the study involved cellular and molecular analyses of isolated PMNs to investigate their responses to CAP treatment. This investigation included assessing migration patterns through metrics such as Track Length (TL) and Track Straightness (TS) and evaluating intracellular ROS production using DHR 123 and NETosis by means of DAPI staining through live cell imaging techniques. Flow cytometry was used to analyze surface antigen expression (CD11b, CD62L, and CD66b), intracellular ROS production (DHR 123), and cell death using propidium iodide (PI) staining in PMNs. An experimental flowchart summarizing these procedures is presented in Figure 2.

### 2.1. Higher RONS Levels in the Upper Respiratory Tract Model with the Plasma Intensive Care Device

ROS and RNS are recognized as critical components of CAP. In initial studies, RONS levels in blood serum and DPBS containing isolated PMNs were determined in the 3D URT model at the positions “throat” and “subglottis” after different treatments (ctrl., PA, PIC, N, NP). Long-lived ROS were quantified by measuring hydrogen peroxide (H_2_O_2_), whereas RNS levels were assessed by detecting nitrite (NO_2−_) and nitrate (NO_3−_). For live cell imaging and flow cytometry experiments, the isolated PMNs were incubated in DPBS, making it essential to understand the accumulation of RONS in this solution and any potential differences resulting from CAP treatment with different devices. Blood serum was additionally chosen for analysis because PMNs were isolated from blood, and complete removal of serum residues during isolation is not feasible. The measurements showed significant stimulation effects for H_2_O_2_ (Figure 3a,d), NO_2−_ (Figure 3b,e), and NO_3−_ (Figure 3c,f) in both blood serum (Figure 3a–c) and isolated PMNs cultivated in DPBS (Figure 3d–f) after PIC treatment. Notably, much higher levels of induction were observed at the “throat” position than at the “subglottis” position. Although there was less increase in RONS after NP treatment, the changes were not statistically significant.

### 2.2. Enhanced Antimicrobial Efficacy in the Upper Respiratory Tract Model with the Plasma Intensive Care Device

The bactericidal effects of the PIC and NP devices were evaluated at the “throat” and “subglottis” measurement points, using *E. coli* as a model organism. At both positions, the PIC device reduced the bacterial count by at least three log_10_ steps (equivalent to 99.9%) after 5 min of CAP application (Figure 4). In contrast, the NP device resulted in a two log_10_ steps reduction (equivalent to 99%) in bacterial count at both positions. These findings indicate that flooding the URT with CAP for 5 min yields better antimicrobial effects with the PIC device than with the NP device. However, with both devices, the antibacterial efficacy is better in the higher “throat” position of the URT model than in the slightly lower “subglottis” position. These results are in accordance with the RONS accumulation results shown in Figure 3 and prove that the amount of RONS correlates with the antibacterial efficacy.

### 2.3. Live Cell Imaging

#### 2.3.1. Impairment of PMN Migration After Treatment with the Plasma Intensive Care Device

The migration behavior of PMNs was assessed by analyzing the migration parameters Track Length (TL) in µm and Track Straightness (TS), which are described in detail in Section 4.5. n = 9 experiments were included, recording n = 5592 tracks in total. During the 20-h observation period, three migration phases were defined: Early Phase, Middle Phase, and Late Phase. The most pronounced differences in PMN migration typically occur during the early hours [30,31]; therefore, only the Early Phase data are shown (Figure 5).

It should be noted that experiments at the “throat” and “subglottis” positions were conducted on different days with different donors, which may have introduced variability due to differences between the donors. However, no significant differences in migration data were observed between the untreated controls. As a result, for clarity and simplicity, these control groups were combined into a single control group (ctrl.).

A significant reduction in TL compared to the untreated control was observed after NP treatment at the “subglottis” position and after PIC treatment at the “throat” position (Figure 5a) with notable median deviations from control. The most pronounced reduction in TL was observed after PIC treatment at the “throat” position, where the median TL was lowest at 100.2 µm, representing just 38% of the TL in the control group (265.2 µm). In contrast, the median TL of PMNs treated with other treatment methods ranged from 212 µm to 291.2 µm. This demonstrates that PMNs treated with PIC at the “throat” position were strongly restricted in their ability to migrate over longer distances.

Similar trends were observed in the analysis of TS (Figure 5b). Both NP treatment at the “subglottis” position and PIC treatment at the “throat” position showed the largest median deviations from the control group. However, only PIC treatment at the “throat” position demonstrated a statistically significant reduction, with a median TS of just 40% in relation to the control (ctrl.).

These findings suggest that PMNs treated with PIC at the “throat” position not only show shorter migration tracks but also a markedly reduced ability to migrate in a directed manner toward the attractant. However, no complete arrest of PMNs was observed in any group. Migration activity persisted across all phases, including the Late Phase, with PMNs continuing to show movement even during the final minutes of the observation period. This indicates that CAP exposure, whether applied with the PIC or the NP device, did not cause “lethal” damage to the PMNs.

#### 2.3.2. No Induction of ROS Production and NETosis of PMNs After Treatment with the Nebulized Air Plasma Device and the Plasma Intensive Care Device

During the 20-h migration observation (Figure 6a), PMNs were also analyzed for intracellular ROS production and NETosis. In the typical progression leading to NETosis, a migrating neutrophil initially slows down, stops, and adopts a rounded morphology (migrating PMNs typically appear irregular due to the extension of pseudopodia). As the cell stops and rounds off, its intracellular ROS concentration reaches a maximum, visible as a red fluorescent signal in live cell imaging (Figure 6b). This process is immediately followed by NETosis, which is characterized by cell membrane rupture and a distinct blue fluorescent signal (Figure 6c) [31,32].

This sequence of events, including the characteristic red or blue fluorescence, was not frequently observed in any of the treatment groups or in the control group (ctrl.). Only sporadic individual cells with no apparent correlation to a specific treatment method showed strong ROS production followed by NETosis, as exemplarily shown in Figure 6. These isolated events, although rare, confirmed the functionality of the applied detection method. Given that these isolated events accounted for only a negligible fraction of the more than 25,000 observed cells and showed no discernible trend, the results indicate that NETosis only occurred at low, random frequency. Similarly, ROS production remained consistent with baseline levels across all treatment groups, with no evidence of increased activity observed in live cell imaging. It should be noted, however, that the quantifiable amount of intracellular ROS production is more accurately measured by means of flow cytometry, as demonstrated in Section 2.4.1.

### 2.4. Flow Cytometry

#### 2.4.1. Reduced Respiratory Burst Potential After Treatment with the Plasma Intensive Care Device

The respiratory burst, an indicator of intracellular ROS production in PMNs, was triggered by fMLP and TNF-α or by PMA after the application of the CAP treatment methods and was subsequently analyzed using flow cytometry with data collected from n = 10 experiments. PMA is membrane-permeable and directly activates intracellular signaling mechanisms, making it a highly potent cell activator that is expected to strongly trigger ROS production, thus serving as a positive control. As described in Section 2.3.1, the untreated samples were combined into a single control group (ctrl.), given the lack of significant differences between these controls across separate experimental days and donors. An Artificial Fluorescence Unit (AFU) was determined based on the fluorescence intensity of Rhodamine 123, which is directly proportional to ROS production and was quantified using flow cytometry. Finally, AFU values were normalized to the control (ctrl. set 1).

The respiratory burst potential of 40% was significantly reduced in the PIC “throat” treatment group in comparison to the ctrl. group. No significant changes were observed in other treatment groups compared to the ctrl. when stimulated with fMLP and TNF-α (Figure 7a).

The positive control with PMA as a stimulant showed the expected effect across all application methods, with PMA triggering the respiratory burst in all groups with no significant differences between the groups compared to the control group (Figure 7b).

The results indicate that the PMNs’ ability to produce ROS is not inherently impaired in any group. This is evidenced by the fact that PMA, which bypasses cell surface receptors and directly activates intracellular pathways, stimulated ROS production equally in all groups. However, this ability was disrupted after PIC treatment at the “throat” position in the case of fMLP and TNF-α, which stimulate ROS production indirectly by binding to cell surface receptors. This finding suggests that while PMNs remain responsive to direct intracellular stimuli such as PMA, their receptor-mediated ROS production in response to bacterial signals (e.g., fMLP) or host-derived inflammatory signals (e.g., TNF-α) is impaired after PIC treatment. However, this decrease in ROS seems to be position-dependent and appears be related to the strong accumulation of RONS at this position in the URT (Figure 3).

#### 2.4.2. Surface Antigen Expression Analysis Shows No Evidence of PMN Activation or Overreaction

The expression levels of the surface antigens CD11b, CD62L, and CD66b on PMNs were quantified by means of flow cytometry to assess cell activation, using data collected from n = 8 experiments. Since treatments at the “throat” and “subglottis” positions were performed on separate days with different donors, significant differences were observed between the untreated control groups due to high donor variability. As a result, the two control groups (designated as ctrl. “throat” and ctrl. “subglottis”) were normalized separately and set as 1 for comparison. Cell activation is typically characterized by an increase in CD11b and CD66b expression along with a decrease in CD62L expression [33,34]. However, no significant differences in the expression of any surface antigen (CD11b, CD62L, and CD66b) were observed in the different treatment groups compared to their corresponding controls (Figure 8). Moreover, no consistent trends suggesting PMN activation were observed. These findings indicate that CAP treatment with either the NP or the PIC device does not provide sufficient stimulation to induce PMN activation, let alone trigger immune overactivation.

#### 2.4.3. No Evidence of Cell Death After Treatment with the Nebulized Air Plasma Device and the Plasma Intensive Care Device

The rate of cell death among PMNs was assessed using propidium iodide (PI) staining to evaluate whether the choice of plasma device or the anatomical position influenced PMN viability, using data collected from n = 10 experiments. All cells detected by means of flow cytometry were classified as either dead or alive. From this classification, a percentage-based proportion in mean values was calculated. The cell death rates showed slight variations at a very low baseline level, ranging from 0.1% to 0.2% (Figure 9). It is important to note that the visualized differences in the diagram appear exaggerated due to the scale of the *y*-axis, which ranges from 0% to 0.5%. This zoomed-in representation highlights even the smallest variations that would otherwise be negligible when using a broader scale (e.g., 0–100%). Therefore, although these differences might seem prominent visually, they are biologically and statistically insignificant. These results align with those presented in Section 2.3.2, demonstrating that the 5-min treatment with either the NP or the PIC device does not cause lethal damage to PMNs.

## 3. Discussion

Previous studies have shown that CAP has strong antibacterial properties without causing cytotoxic damage to the eukaryotic host cells. Notably, no significant side effects have been reported, and no pathogens have been observed to develop resistance to CAP [5,6,7]. These findings, combined with the known ability of CAP to promote wound healing, led to the medical approval of CAP for dermatological applications in 2013 [4,35]. To investigate the application of CAP beyond dermatology, recent research has explored its use in other fields, including the upper respiratory tract (URT). This approach aims to harness the potential of CAP as a safe and effective tool against respiratory infections. However, given the immune responses observed after exposure to CAP in previous studies [17], it is critical to rule out the possibility of an immune overreaction, as such a response could have life-threatening consequences.

To address these concerns, this study investigated the effects of CAP on PMNs in an 3D URT model, focusing on the impact of CAP on the defense mechanisms of PMNs including migration, intracellular reactive oxygen species (ROS) production, NETosis, surface antigen expression, and cell death. Additionally, two CAP devices—the pressurized air plasma (PIC) device and the nebulized air plasma (NP) device, both with distinct CAP modalities—were compared to balance the potent antimicrobial effects of CAP with the need to minimize damage to host immune cells such as PMNs that are crucial for immune defense.

The initial step of the study involved a device-specific comparison to establish baseline information about the PIC and NP devices. RONS measurements showed that the PIC device generated significantly higher amounts of RONS in the treated fluids than the NP device, particularly at the “throat” position. This localized increase is likely due to CAP accumulation caused by the narrower passageway at the vocal cords that restricts airflow and leads to CAP accumulation in the proximal region of the URT. These findings imply that CAP concentrations in human URTs may also vary depending on anatomical structures. In summary, the PIC device appears to carry a higher risk of delivering potentially excessive RONS concentrations to the respiratory tract than the NP device.

The study further examined how these device-specific differences affected antibacterial efficacy. *E. coli*, one of the most frequent pathogens involved in ventilator-associated pneumonia (VAP) [21,36,37], was chosen as a model organism for decolonization. The results showed that both devices reduced the bacterial count by at least two log_10_ steps (equivalent to 99%) at both positions in the URT model after 5 min of treatment. However, the PIC device was slightly more efficacious, achieving three log_10_ steps (equivalent to 99.9%). This increased efficacy is likely due to the higher RONS concentrations produced by the PIC device, as RONS directly attack bacterial membranes, inducing lipid peroxidation and structural damage that ultimately leads to cell death [2,38].

While the PIC device showed greater antibacterial efficacy, it is essential to consider its potential impact on host immune cells. Overly aggressive treatments may impair the body’s natural defenses. To further investigate this issue, the molecular and cell-specific effects of the devices on PMNs were analyzed. When an infection develops within the human immune system, PMNs must first migrate to the affected area before releasing their antimicrobial defenses. This migration, essential for effective pathogen clearance, was assessed after different treatments (ctrl., N, NP, PA, PIC) at the positions (“throat” and “subglottis”) for 5 min, using the parameters Track Length (TL) and Track Straightness (TS). These parameters measure the ability of PMNs to travel significant distances and maintain directional movement toward bacterial chemoattractants such as fMLP. PIC treatment at the “throat” position significantly reduced both TL and TS, indicating impaired PMN migration. Such disruptions may compromise the ability of PMNs to effectively locate and eliminate pathogens, particularly in the first hours after treatment. In contrast, NP treatment caused considerably smaller reductions in TL and TS, indicating that it was less disruptive to PMN migratory function. Although the PIC device achieved superior direct antimicrobial efficacy, its adverse effects on PMN migration may counteract some of these benefits by impairing the immune system’s ability to sustain pathogen clearance over time. This trade-off between immediate bacterial eradication and preserving longer-term immune functionality underscores the need for careful consideration when selecting CAP devices for clinical applications.

An essential focus of this study was to determine whether CAP treatments may induce an immune overreaction in the URT or cause significant cell death among host immune cells. The results indicate that neither the NP nor the PIC device led to a substantial number of cell deaths, suggesting that neither device produces cytotoxic CAP concentrations in the URT after 5 min of treatment. Given the very low maximum cell death rate of 0.2% observed after this exposure time, it is unlikely that cytotoxic effects would occur even after unintentionally extended treatment durations. Naturally, this assumption requires validation through further experimental studies.

Additional insights, important for ruling out an immune overreaction, were provided by the analysis of intracellular ROS production, NETosis, and surface antigen expression analyzed using live cell imaging and flow cytometry.

Upon reaching the site of infection or inflammation, PMNs are capable of killing pathogens through several defense mechanisms [39,40]. A key defenseis the “respiratory burst” that involves a significant increase in oxygen consumption by PMNs [39,41]. During this metabolic process, ROS such as hydrogen peroxide (H_2_O_2_), hydroxyl radicals (OH•), and hypochlorous acid (HOCl) are produced, contributing to pathogen destruction [42].

Another defense mechanism used by neutrophils is NETosis. Neutrophil extracellular traps (NETs) are structures that spread out in a net-like fashion, formed by the dissolution of the nuclear structure while leaving the cell membrane intact. This process involves mixing decondensed chromatin with contents from granules. After the cell membrane ruptures, NETs—composed of DNA from neutrophils, histones, and proteins from azurophilic granules—are released into the extracellular space [33]. In this area, bacteria, viruses, and fungi become immobilized on the NETs, facilitating their phagocytosis by macrophages. Additionally, antimicrobial substances can exert their effects more effectively when pathogens are concentrated at a specific site. NETs also prevent cytotoxic substances released during the immune response from diffusing into surrounding tissues, thereby reducing collateral damage [34].

In addition, the expression of different surface markers play a decisive role during the process of PMN activation. The selectin CD62L is eliminated from the plasma membrane through ectodomain shedding, which in turn is an indicator of neutrophil activation associated with the upregulation of integrin CD11b expression. CD66b, a neutrophil immunoreceptor, also indicates activation by upregulation [43,44].

Despite high extracellular RONS concentrations at the “throat” position after PIC treatment, CAP stimulation did not trigger significant intracelluar ROS production or NETosis. This finding indicates that CAP exposure, even at its highest intensities, was insufficient to activate the defense mechanisms of PMNs in vitro. 

On the contrary, receptor-mediated ROS production triggered by fMLP and TNF-α was even significantly reduced after PIC treatment at the “throat” position. This suggests that high extracellular RONS concentrations may disrupt membrane-bound receptors (e.g., fMLP and TNF-α receptors), impairing their capacity to initiate intracellular ROS production. Still, PMNs remained responsive to PMA, a membrane-permeable ROS trigger, indicating that intracellular signaling pathways were unaffected by CAP treatment. Compared to the untreated control group, surface antigen expression did not show any significant changes after 5 min of either PIC or NP treatment in the different treatment groups. This finding further supported the conclusion that CAP does not induce an immune overreaction under the conditions tested. However, the observed inhibition of receptor-mediated ROS production and the previously discussed impairments in PMN migration after PIC treatment suggest that the effect of PIC may weaken the antimicrobial defense mechanisms of PMNs, potentially counterbalancing its stronger antibacterial efficacy.

This finding raises the broader question of whether the PIC or the NP device is better suited for treating respiratory infections. The pathogens responsible for VAP originate from the unsterile oral flora and move along the endotracheal tube of the patient [19]. From there, they progress deeper into the respiratory tract, ultimately causing pneumonia. If these pathogens primarily travel within biological tissues, one may argue that PMN-mediated defense mechanisms, such as migration within extracellular tissues followed by ROS production, would be essential to combat these infections. However, numerous studies indicate that the pathogens predominantly move along a biofilm on the endotracheal tube, a surface barely accessible to PMNs [19,21]. Therefore, effective disinfection of this non-biological material (e.g., with the PIC device) may be especially beneficial in preventing or managing VAP. Furthermore, the RONS measurements assessed in this study have shown that CAP concentrations significantly decrease in the distal sections of the URT. Additionally, the cuff of the endotracheal tube acts as a barrier, mostly preventing external substances, including CAP, from traveling along the outside of the tube and entering the lung. Thus, it can be concluded that PMNs located in the lung, at the site of infection, would not be functionally impaired after PIC treatment.

Nevertheless, in other scenarios where fully functional PMNs in the URT are essential, a slight reduction in antibacterial efficacy (e.g., the 99% bacterial reduction achieved by the NP device) may be an acceptable trade-off because it preserves full PMN migration and ROS production capabilities. Moreover, a study by Chiappim et al. (2021) [26] showed that the antimicrobial efficacy of nebulized air plasma is highly dependent on the length of the tube through which CAP is applied. This finding suggests that optimizing the tube length from the plasma source to the URT model may potentially further increase treatment efficacy.

Based on this in vitro study, no definitive conclusion can yet be drawn as to which of the two devices would be more suitable for clinical application. The choice is likely to depend on the specific disease and the individual patient characteristics.

Even so, this study marks a significant advancement in establishing the safety of CAP application in the URT, regardless of whether the NP or PIC device is used. These findings lay the foundation for future research, potentially including in vivo models, to further explore the clinical efficacy of these devices. By confirming that CAP application in the URT does not cause lethal damage or an overreaction of the immune system, this study provides a basis for deeper investigation into the clinical potential of CAP in respiratory medicine.

## 4. Materials and Methods

### 4.1. PMN Isolation and Preparation

Blood samples were collected from 15 healthy volunteers, aged 21 to 54 years, using S-Monovette^®^ 7.5 mL LH and Z tubes (Sarstedt AG & Co. KG, Nümbrecht, Germany). Blood serum was separated by density gradient centrifugation at 1700× *g* for 10 min at room temperature using a Heraeus™ Megafuge 1.0 centrifuge (ThermoFisher Scientific, Langenselbold, Germany).

To isolate PMNs, whole blood was carefully layered onto Ficoll-Paque™ PREMIUM (Cytiva Sweden AB, Uppsala, Sweden), a density gradient medium. After approximately 45 min of sedimentation, a layer containing the PMNs, known as the “Buffy Coat”, was formed. The PMNs were then carefully collected from this layer and diluted 1:1 with Dulbecco’s Phosphate Buffered Saline (DPBS, Sigma-Aldrich Chemie GmbH, Steinheim, Germany) for subsequent experimental procedures. This study was approved by the local Ethics Committee of the Medical Faculty of the University of Regensburg, Germany (Approval: [19-1569-101]).

### 4.2. Plasma Devices and Treatment Methods

All experiments were conducted using two prototype devices: the pressurized air plasma (PIC) device and the nebulized air plasma (NP) device (Terraplasma Medical GmbH, Garching, Germany). The PIC device (Figure 1a), the use of which has already been published by Karrer et al. in 2024 [17], is a portable CAP generator connected to a compressor that produces a pressurized airflow (PA) to deliver the generated plasma, referred to as “pressurized air plasma,” into a 3D model of the adult upper respiratory tract (URT). PA alone served as an additional control mode, in which the plasma generator was turned off, allowing only pressurized airflow without plasma. This setup helped distinguish whether any observed cellular changes were due to the pressurized airflow itself or to the combination of additional plasma.

The nebulized air plasma device (Figure 1b) generates CAP with a humidity resistent SMD plasma source and operates in conjunction with an aerosol generator (nebulizer) (Philips InnoSpire Go Mesh nebulizer system; Chichester Business Park, City Fields Way Tangmere, Chichester PO20 2FT UK) that adds NaCl in nebulized form to the plasma stream. While the overall design and plasma generation technology differs from the PIC device, both devices use the same airflow adjustments (0.5 standard liters per minute (slm)) and treatment durations (5 min), ensuring comparable experimental conditions. For additional control, the NP device includes a “nebulized air” (N) mode that applies only nebulized NaCl without plasma, allowing differentiation between effects caused by the plasma mixed with nebulized NaCl versus the nebulized solution alone. As an overall control, PMNs were left completely untreated (ctrl.) to ensure accurate comparability.

The URT model (Figure 1c,d) features two openings at the top, representing the mouth and nose. Through the “mouth” opening, the pressurized air plasma is directed towards deeper regions of the respiratory tract. At the bottom of the model, two slots hold cell culture dishes (35 mm Petri dish; Corning Life Sciences, Wiesbaden, Germany) that serve as placement sites for the samples. These represent anatomical positions within the respiratory tract: one above the vocal cords (position: “throat”) and one below the vocal cords (position: “subglottis”). To simulate the moist environment of the respiratory tract, the interior of the URT model was lined with gauze and rinsed with 25 mL of DPBS immediately before treatment.

### 4.3. Measurement of RONS in Blood Serum and DPBS Containing PMNs

RONS were measured in 35 mm petri dishes (Corning Life Sciences, Wiesbaden, Germany) filled with either 700 µL blood serum or 700 µL DPBS containing isolated PMNs at the position “throat” and “subglottis” in the URT model after different treatments (ctrl., N, NP, PA, PIC). For quantification of H_2_O_2_, a Fluorimetric Hydrogen Peroxide Assay Kit (Sigma Aldrich GmbH, Steinheim, Germany) was used, and fluorescence was measured at an excitation wavelength of 540 nm and an emission wavelength of 590 nm. NO_2_^−^ and NO_3_^−^ concentrations were determined using the colorimetric Nitrite/Nitrate Assay Kit (Sigma Aldrich GmbH, Steinheim, Germany) to detect nitric oxide metabolites at 540 nm absorbance. Fluorescence was measured with a plate reader (Varioscan Flash, Thermo Fisher, Schwerte, Germany), and the assay kits were used as specified by the manufacturer. Each experiment was conducted n = 3, and the results were averaged.

### 4.4. Quantification of Bacteria Inactivation After Treatment with the Nebulized Air Plasma Device and the Plasma Intensive Care Device

A volume of 100 µL of bacteria suspension (∼10^6^/mL; *E. coli* ATCC 25922; LGC Standards GmbH, Wesel, Germany) was applied to Ø 9 cm Müller–Hinton agar plates (Oxoid Deutschland GmbH, Wesel, Germany) and was allowed to dry for 30 min. From Müller–Hinton agar plates, 3 × 35 mm (Ø) samples were punched out, transferred to a 35 mm petri dish (Corning Life Sciences, Wiesbaden, Germany), and subjected to different treatments (ctrl., N, NP, PA, PIC) for 5 min at the “throat” and “subglottis” positions in the URT model. After subsequent incubation of the petri dishes at 37 °C for 24 h, colony forming units (CFUs) were evaluated. For computation of log_10_-reduction rates, CFUs of serial dilutions of the original bacterial suspensions at OD = 0.6 were evaluated. Each experiment was carried out n = 3 in duplicated form, and the results were averaged.

### 4.5. Detection of Cell Migration, ROS Production, and NETosis by Means of Live Cell Imaging

This experiment built on previous studies published by Hundhammer et al. (2022) and Sixt et al. (2023) [45,46]. A chemotaxis assay was conducted to monitor PMN behavior, including migration, ROS production, and NETosis, using 3D µ-Slides (ibidi^®^ GmbH, Martinsried, Germany) in accordance with the manufacturer’s instructions. Each slide consists of three chambers, flanked by reservoirs on both the left and right sides (Figure 10). The chambers were filled with a gel matrix composed of bovine type I collagen (PureCol Bovine Collagen Solution Type I, 3 mg/mL, Advanced BioMatrix, Carlsbad, CA, USA) that was accessible to the PMNs and supplemented with specific fluorescent dyes:Dihydrorhodamine 123 (DHR 123, 10 µM, Thermo Fisher Scientific, Carlsbad, CA, USA) as an indicator of intracellular ROS production;4′,6-Diamidino-2-phenylindole (DAPI, 0.5 µg/mL, Sigma-Aldrich Chemie GmbH, Steinheim, Germany) to visualize extracellular DNA released during NETosis.

After loading the chambers with the gel matrix, they were incubated under humid conditions at 37 °C and 5% CO_2_ for 30 min to allow the gel to solidify. To establish a chemotactic gradient, Formyl-Methionyl-Leucyl-Phenylalanine (fMLP, 100 nM, Sigma-Aldrich Chemie GmbH, Steinheim, Germany) and autologous serum were added to the left reservoir, while isolated PMNs, previously differently treated (ctrl., N, NP, PA, PIC) at the positions (“throat” and “subglottis”), were added into the right reservoirs.

The prepared slides were subsequently placed under the Leica DMi8 inverted live cell imaging microscope (Leica Mikroskopie & Systeme GmbH, Wetzlar, Germany) containing a Leica DFC9000 GT camera with an HC PL FL L20x/0.40 CORR PH1 objective (100x magnification) and illumination provided by the CoolLED pE-4000 light source (CoolLED Ltd., Andover, UK). Phase-contrast and fluorescence images were automatically taken every 45 s with the Leica Application Suite X software version 5.2.2 (Leica Mikroskopie & Systeme GmbH, Wetzlar, Deutschland) over a period of 20 h to track and visualize the migrating PMNs within the channels. Throughout the experiment, samples were maintained in a climate chamber integrated into the microscope stage at 37 °C and 5% CO_2_ to ensure optimal cell culture conditions. 

The images were subsequently analyzed with Imaris 9.0.2 software (Bitplane AG, Zurich, Switzerland). Migration data from the phase-contrast images were captured by analyzing several 30-min intervals within the 20-h imaging period. Depending on the time of acquisition, these intervals were summarized and then categorized into early, middle, and late migration phases. Previous studies have indicated that the most pronounced migration differences tend to occur in the early stages, with fewer differences observed over time [30,31]. Consequently, the observation intervals in the Early Phase were more closely spaced to capture these variations more accurately. Table 1 provides an overview of the time intervals.

Using the “Spots Detection” function of the Imaris software, the tracks of all migrating cells were identified. Program-supported filters (estimated cell diameter, background subtraction, maximum distance, gap size) were applied to minimize interference from artifacts or other cell types. These tracks were then converted into multiple migration parameters by the software, including Track Length (TL) and Track Straightness (TS). TL represents the distance the PMNs have migrated, measured in micrometers (µm). TS is used to measure the linearity of a PMN’s migration path, and has no unit. A perfectly straight path, with no deviation from the direction towards the chemoattractant fMLP would yield a TS of 1. When TS approaches 0, it indicates increasing deviation and a less linear path. Cells with tracks shorter than 25 µm and migration times under 900 s were excluded from the analysis to prevent the inclusion of non-migrating cells or particles in the statistical data. Fluorescence images were used to detect ROS production (red fluorescence) and NETosis (blue fluorescence). PMNs distinguished from the background by displaying clear fluorescence at any point during the observation period were classified as ROS-producing or NETotic cells. This classification was further confirmed when the cells showed the characteristic behavior described in previous studies [31,32] that lead to NETosis: PMNs migrated more slowly (visible in phase-contrast images), stopped, adopted a rounded morphology with increased intracellular ROS concentration, and eventually underwent cell membrane rupture with DNA release. When this distinct sequence of events occurred alongside the fluorescence signals, the cells were confidently classified and counted.

### 4.6. Flow Cytometry Analysis of Cell-Surface Antigen Expression, Respiratory Burst, and Cell Death

Flow cytometry (FACSCalibur, BD Sciences, Franklin Lakes, NJ, USA) was used in combination with the software CellQuest Pro (version 5.2, BD Biosciences) to evaluate the expression of surface antigens (CD11b, CD62L, and CD66b), respiratory burst activity in form of ROS production and cell death rates of PMNs after different treatments (ctrl., N, NP, PA, PIC) at the positions (“throat” and “subglottis”). As in Section 4.5, this approach was based on previous studies [45,46,47].

Surface antigen expression was measured using fluorescently conjugated antibodies: anti-CD11b (ICRF44, PE-conjugated, BioLegend, San Diego, CA, USA), anti-CD62L (DREG-56, FITC-conjugated, BioLegend, San Diego, CA, USA), and anti-CD66b (G10F5, APC-conjugated, BioLegend, San Diego, CA, USA). A volume of 5 µL of each antibody was added to the samples to bind specifically to their corresponding surface epitopes on PMNs. After incubation, the cells were washed to remove any unbound antibodies, enabling accurate measurement of antigen expression by means of flow cytometry.

To quantify ROS production during the respiratory burst, PMNs were incubated in 700 µL DPBS with 10 µL DHR 123 (100 µM, Thermo Fisher Scientific, Carlsbad, CA, USA) and 10 µL seminaphtharhodafluor (SNARF, 10 μM, Life Technologies, Carlsbad, CA, USA). SNARF was included as an internal control without further importance to this experiment. The respiratory burst was triggered using either 10 µL fMLP (10 μM) and 10 µL tumor necrosis factor-alpha (TNF-α, 1 μg/mL, PeproTech by Thermo Fisher Scientific, Carlsbad, CA, USA), or 10 µL phorbol-12-myristate-13-acetate (PMA, 10 μM, Sigma Aldrich Chemie GmbH, Steinheim, Germany) that served as a positive control. For a negative control, no trigger substance was added to the samples. DHR 123 is oxidized in the presence of ROS to Rhodamine 123, a fluorescending compound that can be subsequently quantified by means of flow cytometry. 

To assess the level of PMN cell death, 10 µL propidium iodide (PI, 1.5 µM, Thermo Fisher Scientific, Carlsbad, CA, USA) was added to all samples. PI is a fluorescent dye that penetrates cells with compromised membranes, which is characteristic of dead or dying cells. 

The data were processed and analyzed using FlowJo software (version X 10.0.7r2, FlowJo LLC, Ashland, OR, USA). The quantity of ROS production or surface antigen expression was measured in Artificial Fluorescence Units (AFUs) and was normalized to the corresponding untreated control (ctrl. set 1). AFU is a non-SI unit representing the fluorescence intensity from Rhodamine 123 or the fluorescently conjugated antibodies anti-CD11b, anti-CD62L, and anti-CD66b. As PI-stained dead PMNs show higher fluorescence intensity than living PMNs, they were sorted; subsequently, the software automatically calculated the percentage of dead PMNs relative to total PMNs detected.

### 4.7. Statistical Analysis with GraphPad Prism and SPSS

The statistical analysis of the fluorimetric and colorimetric RONS assays, the analysis of the antimicrobial efficacy and the flow cytometry examinations were conducted using GraphPad Prism software version 10.4.0 (GraphPad Software Inc., San Diego, CA, USA) and are expressed as mean ± standard deviation (± SD) or as mean ± standard error of the mean (SEM) as indicated in the corresponding figure legend. Ordinary one-way ANOVA with Tukey’s multiple comparison test was done to indicate differences of the mean within the ctrl., N, NP, PA, and PIC treatment groups at both positions (“throat” and “subglottis”). Significant results are indicated as * *p* ≤ 0.05, ** *p* < 0.01, *** *p* < 0.001, or **** *p* < 0.0001.

The statistical analysis of the live cell imaging data was performed using the IBM SPSS software version 29.0.0.0 (IBM, Armonk, NY, USA). First, each group was tested for normal distribution using the Kolmogorov–Smirnov test. As all data were normally distributed, simple and grouped box plots were used for graphical representation, and an independent-samples median test was applied to detect significant differences within the samples. Statistical outliers were indicated by circles. Significant results are indicated *** *p* < 0.001.

## 5. Conclusions

This study highlights the potential of cold atmospheric plasma (CAP) as a therapeutic tool for respiratory infections, demonstrating significant differences in the effects of two distinct plasma devices: plasma intensive care (PIC) and nebulized air plasma (NP). The findings indicate that the PIC device, characterized by a higher production of reactive oxygen and nitrogen species (RONS) and enhanced antibacterial efficacy (99.9%) has a more pronounced impact on the functionality and activity of isolated polymorphonuclear leukocytes (PMNs) than the NP device, which produces lower levels of RONS and exhibits weaker antibacterial properties (99%). Given the implications for clinical application in the upper respiratory tract (URT), it is crucial to optimize plasma parameters to ensure maximum safety for patients while preventing immune cell overreaction. Further preclinical studies and advancements in device technology are essential to refine CAP applications and enhance their therapeutic potential in treating respiratory infections such as VAP.

## 6. Patents

WO2022008684A1 (System and plasma for treating and/or preventing a viral, bacterial and/or fungal infection).

## Figures and Tables

**Figure 1 ijms-26-00255-f001:**
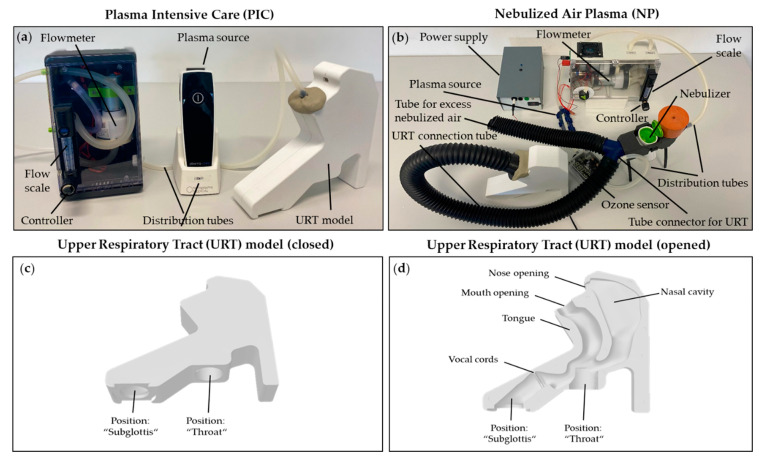
(**a**) The plasma intensive care (PIC) device connected to a 3D model of the upper respiratory tract (URT). Airflow, generated by the flowmeter, is adjustable with the controller between 0.1–1.0 standard liters per minute (slm) and can be monitored on the flow scale. The pressurized air passes through the plasma source (either on or off) and is directed into the URT model through distribution tubes. (**b**) A prototype of the nebulized air plasma (NP) device is shown connected to the URT model, providing an overview of the setup. Airflow, generated using the same setup technology as the PIC device, is directed through two distribution tubes. The left tube channels the airflow, passing it over a humidity resistent SMD plasma source (power supply either on or off) and through an ozone sensor, while the right tube directs airflow through a nebulizer containing NaCl. The contents of both tubes merge at the tube connector before entering a single tube connected to the URT model. A portion of the nebulized NaCl is diverted through an excess tube to prevent over-humidification of the URT model. (**c**) The closed 3D URT model displays the recesses at the “throat” and “subglottis” positions, designed to hold petri dishes containing the samples. The same URT model was used for both devices. (**d**) The opened URT model reveals the anatomical interior of the structure. The tube connecting the devices to the URT model attaches at the mouth opening. The passage for CAP to reach the “subglottis” position is limited at the vocal cord level due to the smaller internal diameter of the conduit.

**Figure 2 ijms-26-00255-f002:**
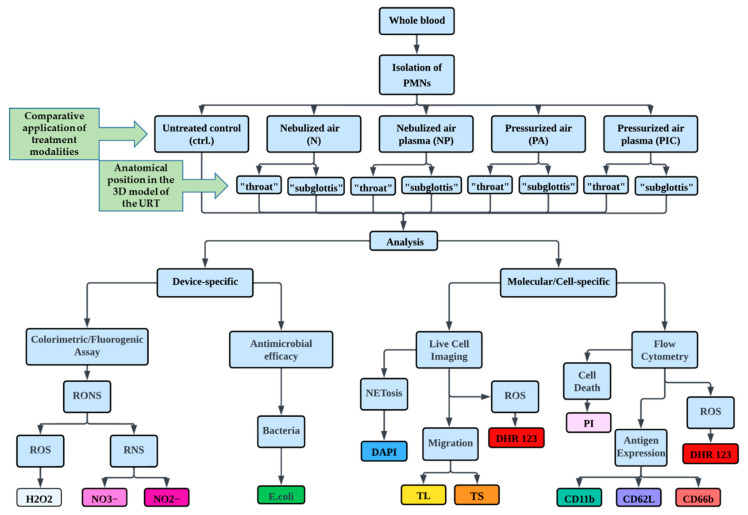
Experimental flowchart of the study. In this in vitro study, polymorphonuclear leukocytes (PMNs) were isolated from whole blood and subjected to various treatments for 5 min. The treatments included nebulized air (N), nebulized air plasma (NP), pressurized air (PA), pressurized air plasma (PIC), and a control group (ctrl.) without treatment. The applications were administered at two specific anatomical positions within a 3D URT model: the “throat” and the “subglottis”. Device-specific investigations were conducted to assess the accumulation of reactive oxygen and nitrogen species (RONS) in the fluids used during the experiments, such as blood serum and DPBS containing isolated PMNs, and to evaluate the antibacterial efficacy of the NP and PIC devices using *E. coli* as a model organism. Subsequent cell biological analyses of the PMNs involved live cell imaging techniques to assess intracellular reactive oxygen species (ROS) using DHR 123, quantify NETosis through DAPI staining, and evaluate cell migration by measuring Track Length (TL) and Track Straightness (TS). In addition, flow cytometry was used to investigate molecular changes in PMNs, focusing on surface antigen expression levels of CD11b, CD62L, and CD66b, as well as on intracellular ROS intensity and cell death using propidium iodide (PI) staining.

**Figure 3 ijms-26-00255-f003:**
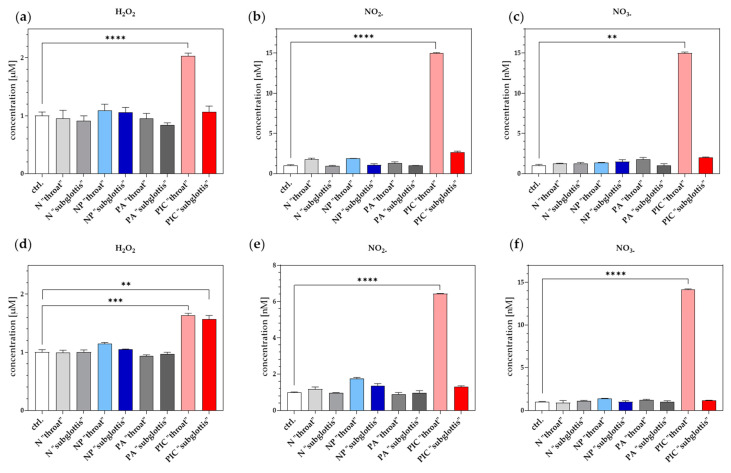
H_2_O_2_, NO_2−_, and NO_3−_ in blood serum and isolated PMNs maintained in DPBS. (**a**–**c**) A volume of 700 µL of blood serum was treated (ctrl., PA, PIC, N, NP) at both positions (“throat” and “subglottis”) for 5 min (n = 3). (**d**–**f**) Isolated PMNs in 700 µL DPBS were treated accordingly. (**a**,**d**) Using a H_2_O_2_ standard series, a Fluorimetric Hydrogen Peroxide Assay Kit was used to determine H_2_O_2_ concentration [µM] in the fluids. (**b**,**e**) NO_2−_ and (**c**,**f**) NO_3−_ concentration [nM] in fluids were quantified using a colorimetric Nitrite/Nitrate Assay Kit. Statistical analysis: Ordinary one-way ANOVA with Tukey’s multiple comparison test was done to compare the mean of untreated ctrl. to all treatments at both positions. ** *p* < 0.01, *** *p* < 0.001, **** *p* < 0.0001.

**Figure 4 ijms-26-00255-f004:**
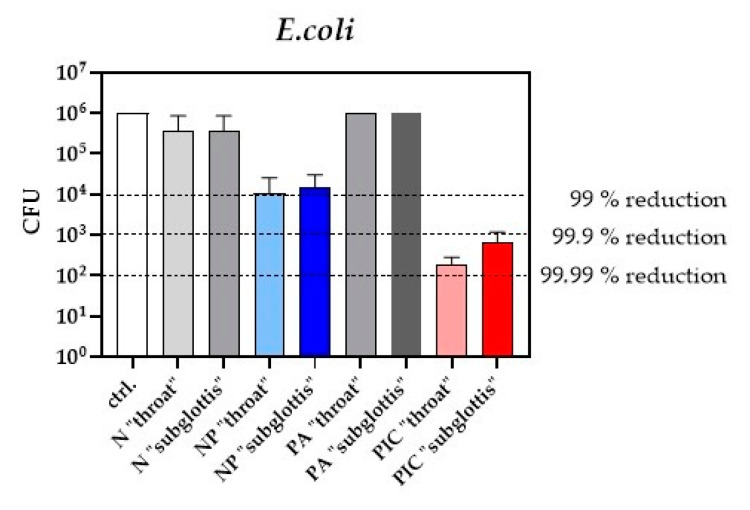
Decolonization of *E. coli* using the NP and the PIC device. Colony Forming Units (CFUs) were determined after different treatments (ctrl., N, NP, PA, PIC) at the “throat” and “subglottis” positions in the 3D URT model. Black dotted lines indicate the reduction of two log_10_ steps (99%), three log_10_ steps (99.9%), and four log_10_ steps (99.99%) of viable bacteria. (n = 3, mean ± SD).

**Figure 5 ijms-26-00255-f005:**
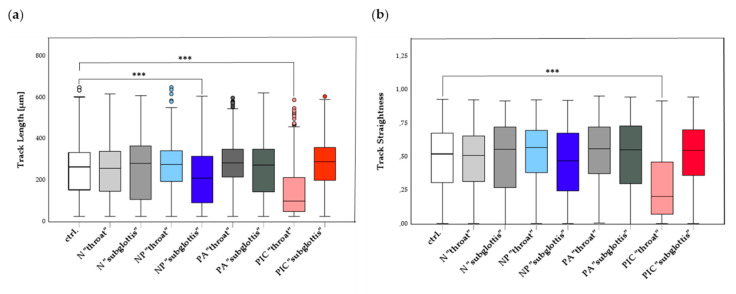
PMN migration was analyzed using live cell imaging. (**a**) Early Phase Track Length (TL) and (**b**) Track Straightness (TS) of isolated PMNs was determined after treatments (ctrl., N, NP, PA, PIC) at the positions “throat” and “subglottis” for 5 min (n = 9). Both parameters were evaluated by means of live cell imaging. Statistical analysis: An independent-samples median test was done to compare the median of the untreated ctrl. to all remaining treatments at both positions. Statistical outliers are represented as circles. *** *p* < 0.001.

**Figure 6 ijms-26-00255-f006:**

Original optical images as an exemplary presentation of intracellular ROS production and NETosis of PMNs during the migration process, analyzed using live cell imaging. Yellow arrows indicate the same PMN across all images. (**a**) Phase-contrast image showing migrating PMNs, some with clearly visible pseudopodia (blue arrows). The yellow arrow highlights a PMN that has stopped migrating and adopted a rounded morphology. (**b**) The same PMN, now stationary, shows increased intracellular ROS concentration, visible as a strong red fluorescent signal. A faint red glow is also discernible in other PMNs (purple arrows), reflecting the baseline intracellular ROS concentration present in all PMNs. (**c**) After cell rupture and the release of cellular DNA, the PMN displays a distinct blue fluorescent signal, indicating NETosis. No other cells in this image show extracellular DNA release, resulting in a single visible blue signal.

**Figure 7 ijms-26-00255-f007:**
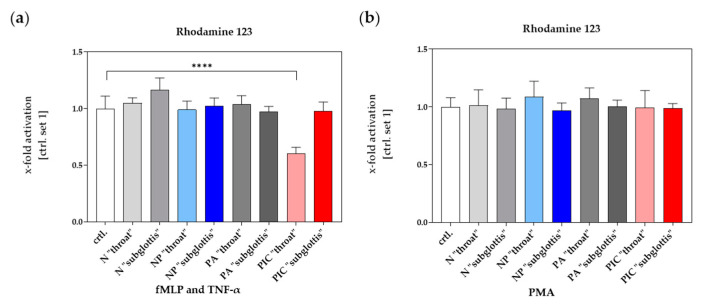
ROS production of isolated PMNs after different treatments (ctrl., N, NP, PA, PIC) at the positions (“throat” and “subglottis”) for 5 min (n = 10, mean ± SEM) and triggered by either fMLP and TNF-α (**a**) or PMA (**b**). The quantification by means of flow cytometry of Rhodamine 123 as an indicator of intracellular ROS production in PMNs was calculated in Artificial Fluorescence Units (AFUs) and finally normalized to the untreated control group (ctrl. set 1). Statistical analysis: Ordinary one-way ANOVA with Tukey’s multiple comparison test was used to compare the normalized untreated ctrl. to all treatments at both positions. **** *p* < 0.0001.

**Figure 8 ijms-26-00255-f008:**
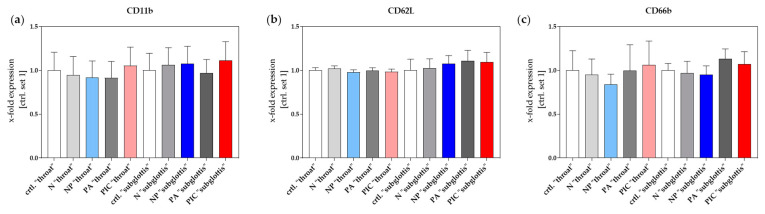
Expression levels of surface antigens CD11b (**a**), CD62L (**b**), and CD66b (**c**) of isolated PMNs after different treatments (ctrl., N, NP, PA, PIC) at the positions “throat” and “subglottis” for 5 min (n = 8, mean ± SEM). The quantification of fluorochrome-conjugated antibodies by means of flow cytometry as an indicator of individual surface antigen expression in PMNs was performed in Artificial Fluorescence Units (AFUs) and normalized to the control (ctrl. set 1). Statistical analysis: Ordinary one-way ANOVA with Tukey’s multiple comparison test was used to compare the treatment groups to the corresponding normalized untreated controls. No significant differences were observed between the individual treatments and the corresponding control group.

**Figure 9 ijms-26-00255-f009:**
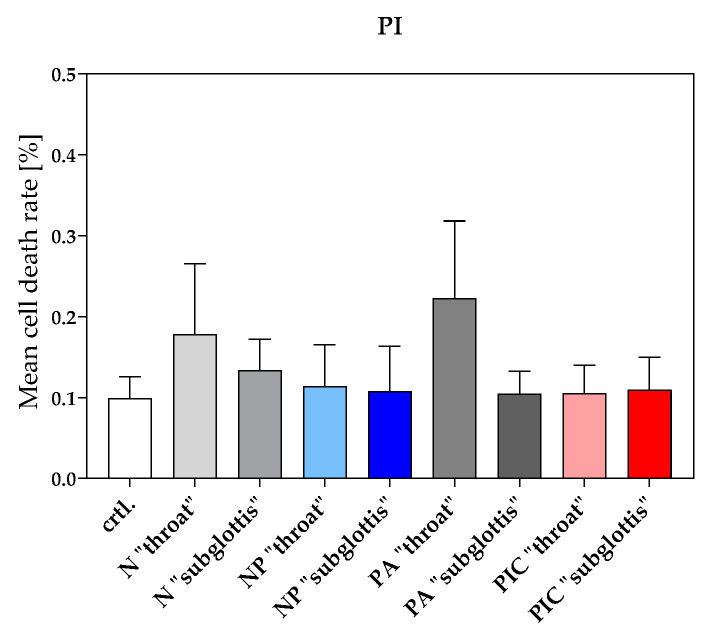
Mean cell death rate [%] of PMNs after different treatments (ctrl., N, NP, PA, PIC) at the positions “throat” and “subglottis” for 5 min (n = 10, mean ± SEM). Quantification of dead cells was performed using flow cytometry after PI staining and is presented as the percentage of dead PMNs relative to the total PMNs detected within each treatment group. Statistical analysis: Ordinary one-way ANOVA with Tukey’s multiple comparison test was used to compare the treatment groups to the untreated control (ctrl.). No significant differences were observed between the individual treatment groups and the untreated control (ctrl.).

**Figure 10 ijms-26-00255-f010:**
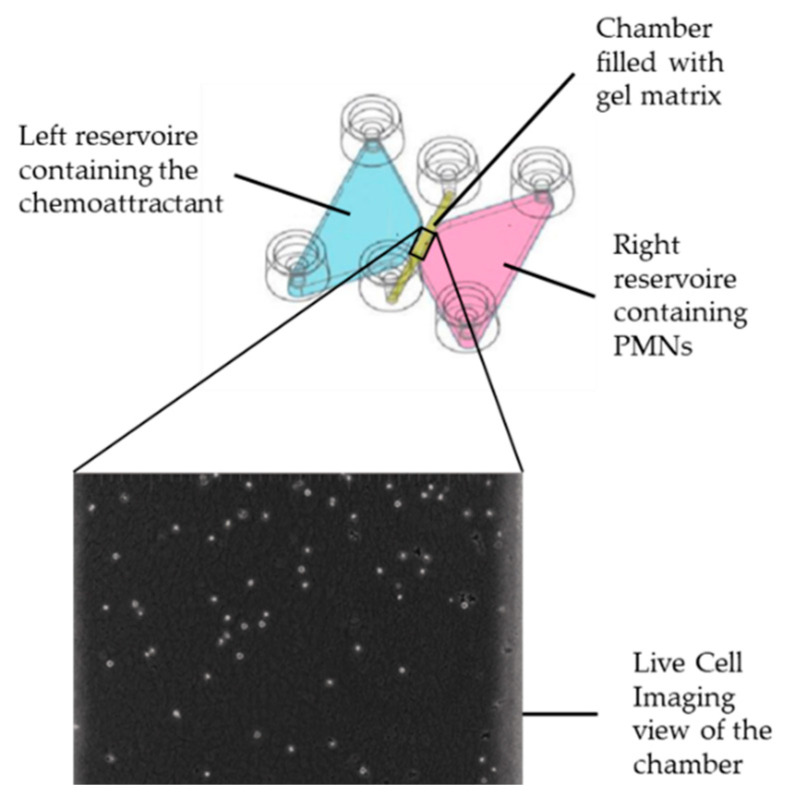
3D µ-slides contain three chambers, each flanked by reservoirs on the left and right side. The upper section of the image provides a schematic representation of one-chamber setup. PMNs are pipetted into the right reservoir and migrate through the chambers filled with gel matrix to reach the chemoattractant fMLP in the left reservoir. Throughout the observation period, the live cell imaging camera remains focused on the chambers. The lower section of the image shows a representative enlarged snapshot of a chamber, with individual grey dots representing PMNs.

**Table 1 ijms-26-00255-t001:** The Early, Middle, and Late Phases of PMN migration; each consists of three summarized 30-min intervals in which PMN migration was analyzed.

Early Phase	Middle Phase	Late Phase
minute 30–60	minute 150–180	minute 600–630
minute 60–90	minute 240–270	minute 960–990
minute 90–120	minute 330–360	minute 1125–1155

## Data Availability

The data presented in this study are available on request from the corresponding author.

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
