# Peer review of "Cellular Response of Immune Cells in the Upper Respiratory Tract After Treatment with Cold Atmospheric Plasma In Vitro"

_ijms, 2024, doi:10.3390/ijms26010255_

Round 1

Reviewer 1 Report

Comments and Suggestions for Authors

The research quality and clarity of the arguments are truly remarkable. This seems good work, and I believe it makes valuable contributions to the field. However, I would like to offer some small observations that could further improve your manuscript.

Regarding the arrangement of figures in your article, Figure 1 is at the end, while subsequent figures begin with Figure 2. The figures must be sequentially from Figure 1 to the last to facilitate understanding and following the content.

I have reviewed your work and can say that it is well-structured. However, I feel the English needs to be revised as the writing has some irregularities. For example, line 67 mentions "... both in in vitro ..." where there is unnecessary repetition.

In the description of Figure 1, the authors mention that the tubes carrying the airflow and the CAP-treated nebulizer are connected to enter the URT model through a single tube. Including this information in the figure would help to provide a complete diagram. Furthermore, including information about plasma reactors in the present work might be beneficial, as it would enrich the reader’s knowledge and understanding of the experimental setup.

Comments on the Quality of English Language

It would be highly recommended that the document be reviewed.

Reviewer 2 Report

Comments and Suggestions for Authors

Leonardo Zamorano Reichold, et al. Investigated the cellular response of immune cells in the upper respiratory tract after CAP treatment.

I have the following suggestions before an acceptance.

[1] Please provide the original PMN migration optical images in Figure 5.

[2] Please provide scar bars in Figure 6.

[3] ROS signal shown in Figure 6 is weak. Do you choose the right ROS probe to do ROS measurement? In plasma medicine, most scholars use DCFDA-based probes. I recommend the authors try different but stronger ROS probes to do measurements.

[4] Please provide the original ROS flow cytometry data in Figure 8. 

Reviewer 3 Report

Comments and Suggestions for Authors

Dear Authors,

The manuscript entitled 'Cellular response of immune cells in the upper respiratory tract after treatment with cold atmospheric plasma in vitro' presents experimental results obtained from the interaction of a plasma source with cells. The authors present the data obtained from the point of view of the antimicrobial action of plasma, in the two proposed arrangements, namely: pressurized air plasma with the plasma intensive care (PIC) device, as well as nebulized air plasma (NP) with a new humidity resistant surface-microdischarge (SMD) plasma source.

The text of the manuscript is well-written, structured and organized, each paragraph being supported by the necessary explanations, references and experimental results. The figures are of high quality, suitable for journal standards. The conclusions are also well-structured and succinctly present the important results of the study presented by the authors.

After a review of the entire manuscript by the editorial team (to avoid possible writing / typing errors) I recommend this manuscript for publication in its present form.

Reviewer 4 Report

Comments and Suggestions for Authors

The topic addressed by the authors - CAP, is a complex one and emphasizes the importance of alternative procedures in treating certain pathologies, such as upper respiratory tract infections. In the current context, multidrug resistance to antibiotics represents a global burden, and there is an urgent need for discovering an innovative treatment method. Thus, CAP has been well studied in terms of dermatological conditions and has gained significant attention in recent years due to its antimicrobial properties and its potential to modulate the immune response. The article discusses, in an original way, a current topic and opens new perspectives for research.

The subject discussed in this article is important for the scientific medical field, and I believe that both the review of what we know so far about CAP and the technique used to discover potential effects of CAP are extremely well-organized.

The comparison between the two techniques, CAP – NP 114 and PIC, in terms of evaluating their efficacy and safety for further use in the treatment or prevention of respiratory infections (VAP), is rigorously presented. The fact that theoretical data and results are presented in a diagrammatic form helps the reader understand the information more easily, and the presentation of the devices used - pictures is a good way to make the reader feel as though they are actually in the laboratory working on this project.

The English presentation is very good, and going through the material was not problematic from a linguistic perspective.

The research is credible, and the bacteriological evidence presented is analyzed according to standards, with the results being adequately argued. Compared to other studies related to CAP, this one raises the possibility of using the method for other pathologies, not just dermatological ones.

I like that the material is well-structured, the language used is clear, the information is presented concisely, and the bibliographic references are adequate and related to the objectives of the study because they strengthen the arguments made in the article and the authors shows that their claims are grounded in evidence.

In conclusion, I think this paper is excellent and is an important addition to the medical literature.
